# Facile Synthesis of Ag Nanowire/TiO_2_ and Ag Nanowire/TiO_2_/GO Nanocomposites for Photocatalytic Degradation of Rhodamine B

**DOI:** 10.3390/ma14040763

**Published:** 2021-02-06

**Authors:** Pejman Hajipour, Abbas Bahrami, Maryam Yazdan Mehr, Willem Dirk van Driel, Kouchi Zhang

**Affiliations:** 1Department of Materials Engineering, Isfahan University of Technology, Isfahan 84156-83111, Iran; p.hajipourm@gmail.com (P.H.); a.n.bahrami@gmail.com (A.B.); 2Faculty EEMCS, Delft University of Technology, Mekelweg 4, 2628 CD Delft, The Netherlands; willem.dirk.van.driel@signify.com (W.D.v.D.); G.Q.Zhang@tudelft.nl (K.Z.)

**Keywords:** photocatalytic materials, Ag nanowire, surface plasmon resonance, nanocomposite, Rhodamine B

## Abstract

This paper investigates the photocatalytic characteristics of Ag nanowire (AgNW)/TiO_2_ and AgNW/TiO_2_/graphene oxide (GO) nanocomposites. Samples were synthesized by the direct coating of TiO_2_ particles on the surface of silver nanowires. As-prepared AgNW/TiO_2_ and AgNW/TiO_2_/GO nanocomposites were characterized by electron microscopy, X-ray diffraction, UV/visible absorption spectroscopy, and infrared spectroscopy. Transmission electron microscope (TEM) images confirmed the successful deposition of TiO_2_ nanoparticles on the surface of AgNWs. The photocatalytic activity of synthesized nanocomposites was evaluated using Rhodamine B (RhB) in an aqueous solution as the model organic dye. Results showed that synthesized AgNW/TiO_2_/GO nanocomposite has superior photocatalytic activities when it comes to the decomposition of RhB.

## 1. Introduction

Industrial developments, expansion of cities, and worldwide population growth have put pressure on available water resources. Drinking water safety has become a pressing issue worldwide, especially in developing countries. Millions of people around the world have limited access to a safe drinking water source and are threatened by water-borne diseases annually [1].

Sustainable development solely based on natural water resources is certainly not a possibility anymore. Wastewater recycling is, therefore, an inevitable choice. Wastewater treatment is a complex multicomponent process. Removal and decomposition of toxic agents/components and pollutants necessitate the use of highly functional and efficient methods. Traditionally, there are different chemical and physical processes for wastewater treatment. Chemical processes are based on chemicals such as chlorine, ozone, and chlorine dioxide, whereas physical treatments are based on filtration of water, thermal processes, and UV radiation methods. A functional wastewater treatment system should primarily be capable of bringing water back to the consumption cycle.

Among different advanced highly functional wastewater treatment techniques, photocatalyst materials have recently emerged as a promising method for wastewater treatment [2,3,4,5]. Photocatalysts are defined as a class of materials capable of producing transformations in chemical reaction partners upon light absorption. These transformations happen as a result of electron–hole pair formation. The energy, required to create electron–hole pairs, is provided by light radiation, which is often UV light [6,7]. The photo-generated electrons react with O_2_, resulting in the generation of active anion radicals, such as O^•2−^. The holes, on the other hand, generate active OH^•^ radicals. Produced active radicals decompose organic compounds, including oil, fat, and dyes. Photocatalysts are very promising when it comes to the decomposition and removal of organic pollutants and microorganisms in wastewater [5,8]. As it has been shown, photocatalytic treatment of wastewater can also eliminate loads of microbiological contaminations that may transmit infectious disease [9,10]. Among different photocatalyst compounds, TiO_2_ is the first and still one of the most widely used ones. A significant drawback with TiO_2_ is its relatively high bandgap, which is a considerable restriction when it comes to making full use of sunlight [11,12]. Additionally, the low surface-to-volume ratio in TiO_2_ can also adversely influence the efficiency of photocatalytic oxidation reactions.

Ag nanostructures, especially Ag nanowires, exhibit exceptional activities in catalytic, chemical, and biological applications [13]. This is due to their unique chemical, photochemical, and electrical properties, including superior charge mobility. More importantly, Ag nanowires can immobilize TiO_2_ nanoparticles. This is extremely important when it comes to the fixation of TiO_2_ nanoparticles inside a photoreactor. Additionally, Ag nanowires can act as a surface plasmon resonance (SPR) photosensitizer, which in turn reportedly increases photocurrent density [13]. Graphene oxide (GO) is also known to have extraordinary adsorption properties, helping to increase the concentration of organic pollutants on the surface of hybrid photocatalyst composite and, therefore, to increase reaction sites [14,15,16,17,18]. GO can also act as an electron reservoir, resulting in faster electron conduction of electron–hole pairs towards organic contaminants.

Rhodamine B (RhB) is a widely used xanthene dye with various applications in the textile, paint, paper, leather, and pharmaceutical industries. As RhB can irritate the skin and eyes and also cause problems for the alimentary tract and the respiratory system, the proper degradation of this dye in industrial wastewater is very important before introducing the wastewater into ecological systems [19]. The goal of this study is to develop a facile synthesis process for a highly efficient and functional TiO_2_-based photocatalytic structure, making use of the non-toxicity and applicability of TiO_2_ while addressing its limitations through nanocomposite formation. The plasmonic effects of AgNWs and charge separation properties of GO are utilized and studied for this purpose. To this end, TiO_2_ nanoparticles are synthesized on Ag nanowires, and the effects of the presence of GO nanosheets on structural and photocatalytic properties of the synthesized materials are evaluated. Finally, a possible photocatalytic reaction mechanism and degradation process is discussed.

## 2. Materials and Methods

### 2.1. Material and Chemicals

Graphene oxide (GO) powder (99.99%) was purchased from Pishgaman NanoMavad (Mashhad, Iran). GO powder specifications, provided by the supplier, are summarized in Table 1. Ethylene glycol (C_2_H_6_O_2_, 94.5%), Polyvinylpyrrolidone (PVP, average Mw: ~55,000), M CuCl_2_, AgNO_3_, and sulfuric acid (H_2_SO_4_, 98.1%) were all purchased from Merk (Ann Arbor, MI, USA)

### 2.2. Synthesis of AgNW/TiO_2_ and AgNW/TiO_2_/GO Nanocomposites

Ag nanowires were synthesized based on the polyol method reported in the literature [20]. In this research, 0.4 g Polyvinylpyrrolidone (PVP) was dissolved in 10 mL ethylene glycol (EG) in a beaker by vigorous stirring. After complete dissolution, 0.01 M CuCl_2_ was dissolved in EG and two milliliters of this solution was added into the PVP/EG solution. Then, 0.2 g of AgNO_3_ was added to the EG, PVP, and CuCl_2_ mixture. The mixture was then kept at 160 °C for 40 min. The solution was then naturally cooled down to RT, followed by centrifugation. Ag nanowires, obtained by centrifugation, were washed with ethanol/de-ionized water to make sure that there was no EG left with nanowires. To synthesize AgNW/TiO_2_ and AgNW/TiO_2_/GO nanocomposites, 0.1 g purified AgNWs was dispersed in 25 mL of 1:1 water/ethanol solution. Then, 74.7 μL of TTIP was added to the mixture, followed by stirring at room temperature (for 24 h). The stirring at this stage was conducted to make sure that a complete hydrolysis/condensation of TTIP was attained. The pH of the mixture was monitored and kept within 6–7 throughout the synthesis, using 0.1 M NaOH/HCl solutions. AgNW/TiO_2_ nanocomposite was then attained by adding commercial TiO_2_ nanoparticles, while the mixture was centrifuged, followed by 3 times washing with ethanol to make sure that there was no excess TTIP left in final samples. The same process was performed for the second vial, and in this case, 0.01g of GO was added to the mixture. Figure 1 shows the schematics of the mentioned synthesis routes.

### 2.3. Characterization

Synthesized nanocomposites were characterized using both a transmission electron microscope (TEM, JEOL JSM-6710F, Pleasanton, CA, USA) and X-ray diffraction (XRD) with CuK_α_ radiation (λ = 1.54 Å). Characteristic plasmon resonance of Ag, AgNW/TiO_2,_ and AgNW/TiO_2_/GO nanocomposite samples was studied by a Univkon-XL UV–VIS-NIR scanning spectrophotometer (OR, USA) within 200–1100 nm. Fourier transformed infrared (FTIR) analysis was carried out using a Perkin–Elmer Spectrum (Princeton, NJ, USA) 100 series for 200 scans at a resolution of 4 cm^−1^. The photocatalytic characteristics of synthesized nanocomposite samples were evaluated in Rhodamine B (RhB)-containing wastewater. RhB is a well-known contaminant in the textile, plastic, and dye industries. Releasing RhB-containing wastewater into nature is associated with severe long-term adverse effects on the environment.

### 2.4. Photocatalytic Experiments

A 12 mg/L RhB/de-ionized water solution was used to assess the photocatalytic response of TiO_2_, AgNW/TiO_2_, and AgNW/TiO_2_/GO nanocomposite samples. RhB solution was added to three quartz cuvettes. Then, 30 mg from each specimen (TiO_2_, AgNW/TiO_2_, and AgNW/TiO_2_/GO nanocomposites) was mixed with RhB-containing solution. The contents of photocatalyst samples in all solutions were similar (12 mg/L). Solutions were then slowly stirred for 2 h in a dark chamber to make sure that photocatalysts powders and RhB were very well mixed and were in equilibrium. After 2 h stirring, samples were exposed to light under the Xenon light source of 75.9 KJ/ m^2^. While exposed to light, samples were constantly stirred. Figure 2 illustrates the schematic diagram of the photocatalytic reactor.

## 3. Results and Discussion

### 3.1. Characterization of Synthesized Nanocomposites

Figure 3 illustrates the XRD patterns of the purchased GO nanopowder and synthesized AgNW, AgNW/TiO_2_, and AgNW/TiO_2_/GO nanocomposites. Figure 3b shows the XRD pattern of commercial P25 TiO_2_ nanoparticles comprising both rutile and anatase phases. The anatase and rutile phases are in good agreement with the JCPDS (Card No: 96-720-6076 for anatase and Card No: 96-900-4143 for rutile phases). The diffraction peaks in Figure 3c are very well matched with those of cubic silver in accordance with the JCPDS (Card No: 96-900-8460), inferring that AgNW was successfully synthesized. The diffraction peaks at *2θ* = 38.1, 44.2, 64.4, and 77.4° correspond to (111), (200), (220), and (311) crystallographic planes of Ag crystal, respectively. The XRD patterns of AgNW/TiO_2_ and AgNW/TiO_2_/GO samples were found to be similar to that of AgNW, except for the peak at 25.2°, which is attributable to titania crystal. The characteristic 25.2° peak in synthesized AgNW/TiO_2_ and AgNW/TiO_2_/GO nanocomposites, shown in Figure 3d,e, respectively, indicates the successful formation of anatase TiO_2_ phase on AgNW structures. The absence of Ag_2_O-related peaks in XRD spectra indicates that Ag nanowires were not oxidized during the reaction. The fact that no peak, related to the GO, is observed in XRD spectra is attributable to the fact that the GO concentration in AgNW/TiO_2_/GO nanocomposite is below the detection limit of the XRD technique.

Figure 4 shows TEM images of the synthesized AgNW, AgNW/TiO_2_, and AgNW/TiO_2_/GO samples. Results showed that Ag nanowires with an average length of 3 µm and an average diameter of 50 nm were successfully synthesized. Figure 4a shows an example of a Ag nanowire. Figure 4b shows that TiO_2_ particles are very well attached to the surface of AgNWs, which is very well in line with the idea of having firmly coupled AgNW/TiO_2_ entities. This ensures that TiO_2_ nanoparticles are very well kept at their sites during photocatalytic reactions. Figure 4c shows how GO sheets are distributed inside the AgNW/TiO_2_/GO nanocomposite sample. Figure 4d shows size of P25 TiO_2_, and Figure 4e illustrates GO samples (image of GO is provided by the supplier,.Some clusters of TiO_2_ nanoparticles appear to be attached to GO sheets. This enhances electron mobility inside the composite, and this, in turn, improves the photocatalytic efficiency of the synthesized composite. Even though GO is electrically insulating due to a disruptive sp2 bonding network, it can become conductive by restoring the network. More importantly, GO sheets are expected to significantly enhance the adsorption capabilities in synthesized samples, due to the large surface-to-volume ratios in GO. In that sense, GO sheets can be seen as entrapment sites for contaminants. This is a critical point in improving the efficiency of photocatalytic reactions.

Figure 5 shows the UV-Vis spectra of both AgNW/TiO_2_ and AgNW/TiO_2_/GO samples. It is seen that upon the integration of Ag nanowires, the optical absorption spectrum shows a rather significant red shift towards the visible region. Additionally, such improvement in the optical absorption of the AgNW/TiO_2_ sample is also possibly attributable to the effects of Ag nanowires on the dielectric constant of the surrounding mixture. When GO was added to nanocomposites, the optical absorption was further increased within the visible light region. This can be taken as an indication that the GO-containing sample has a lower optical bandgap. It is known that there are many unpaired π-electrons inside GO, due to its lamellar structure. This can result in the binding of free electrons to the surface of TiO_2_, which in turn results in the formation of Ti‒O‒C bonds. This reportedly leads to an upward shift of the valance band edge, which ultimately results in a reduced bandgap [11,12]. Figure 6 shows the Tauc plot of the modified Kubelka–Munk (KM) function with a linear extrapolation used to calculate the direct bandgap values of P25 TiO_2_, AgNW/TiO_2_, and AgNW/TiO_2_/GO materials, i.e., 3.46, 3.11, and 3.00 eV, respectively.

Figure 7 depicts the FTIR spectra of the AgNW/TiO_2_ and AgNW/TiO_2_/GO nanocomposites, as well as AgNW, P25 TiO_2_, and GO samples. The peaks near 1400 and 650 cm^−1^ are ascribed to Ti–O–Ti stretching, relating to lattice vibrations of TiO_2_. Peaks, observed at 3350 and 1627 cm^−1^, are assigned to the stretching and bending modes in OH. This indicates that some OH groups are absorbed over the surface [12]. Only these two peaks could be detected in AgNWs samples, as expected of a metallic material. The addition of Ag nanowires resulted in a slight shift in the TiO_2_ lattice vibration towards 1335 cm^−1^. This confirms that Ag–TiO_2_ bonds formed within the composite. Peaks observed at 3400, 2900, 1975, 1637, 1570, 1375, 1220, and 850–1200 cm^−1^ are assigned to O–H, CH_2_, –COOH, C=O, C=C, COO, C–OH, and C–O functional groups, respectively [21]. The GO-containing sample showed some extra absorption peaks within the IR range, more specifically 1065, 1155, 1225, and 1375 cm^−1^. These peaks can be ascribed to C–O, C–OH, and COO–stretching in GO. The intensities of the mentioned GO-related peaks are rather weak, which possibly has to do with the possible compound formation between GO and TiO_2_ [12].

### 3.2. Characterization of Photocatalytic Behaviour of Nanocomposites

Photocatalytic reactions, occurring at the surface of the TiO_2_ nanoparticles, result in the formation of free radicals, as follows:TiO_2_ + hν → h^+^_VB_ + e^−^_CB_(1)
e^−^_CB_ + O_2_ → O^•^_2_^−^(2)
O^•^_2_^−^ + H^+^ → HOO^•^(3)
h^+^_VB_ + H_2_O → OH^•^ + H^+^(4)
RhB + {O^•^_2_^−^, HOO^•^ or OH^•^} → OOH^−^ or OH^−^ intermediates → products(5)

In this process, photogenerated e^−^ in the conduction band of the TiO_2_ catalyst moves toward the surface and is scavenged by the O_2_ present in the solution, resulting in the formation of superoxide anions (O^•^_2_^−^). Simultaneous protonation of the generated O^•^_2_^−^ results in the formation of hydroperoxyl radical (HOO^•^). h^+^ present in the valance band of the TiO_2_ would react with H_2_O molecules, resulting in the creation of hydroxyl radical (OH^•^). Produced active species would then react with RhB, ultimately resulting in wastewater treatment. Time-dependent UV-visible absorption spectra of the RhB/ AgNW/TiO_2_/GO solution sample are shown in Figure 8.

The absorption spectra of three other catalysts samples are the same as this one with different absorbance. The photocatalytic activities of AgNW, AgNW/TiO_2_, and AgNW /TiO_2_/GO samples were evaluated under Xenon light radiation. Figure 9 shows the changes in relative concentration (C/C0) of RhB. It was found that the Ag-containing GO/TiO_2_ sample showed the highest degradation efficiency. One can, therefore, conclude that the superior photocatalytic performance of AgNW/TiO_2_/GO nanocomposite is due to combinatorial contributions of TiO_2_ and Ag/GO networks. AgNW/TiO_2_/GO showed the degradation of RhB with a rate constant of 5.54 × 10^−2^ min^−1^, which is two times faster than the photocatalytic activity of AgNW/TiO_2_.

Figure 10 shows the kinetics of RHB degradation. The reaction rate constant was calculated using the below equation:Ln(C/C_0_) = −*K*t(6)
with C being the instant RhB concentration, C_0_ the initial RhB concentration, *K* the pseudo first-order rate constant, and t the reaction time in minutes. The values for the degradation rate constant were calculated to be 0.054, 0.026, 0.011, and 0.0003 min^−1^ for AgNW/TiO_2_/GO, AgNW/TiO_2_, P25 TiO_2_, and Ag samples, respectively.

Compared to TiO_2_ nanoparticles, AgNW/TiO_2_ nanocomposites display improved photocatalytic properties because they exhibited a perfect electron exchange between the photoexcited TiO_2_ and Ag particles [13,22]. Spherical silver nanoparticles are expected to have lower photocatalytic efficiency, given that trapping sites on the surface of spherical particles further stimulates the e^−^/h^+^ recombination reactions. One-dimensional AgNWs decrease the charge localization over the surface, associated with reduced e^−^/h^+^ recombination. Reduced e^−^/h^+^ recombination has positive implications for the photocatalytic activities of nanocomposite samples [22]. This improvement can be ascribed to the localized SPR effect of Ag nanowires. In this phenomenon, when AgNW is irradiated with light, the charge density is redistributed, which creates a strong coulombic restoring force resulting in the oscillation of charge density in phase with the incident light [23,24,25]. Figure 11b shows this effect for AgNWs. This oscillation would then distress the dielectric constant of the surrounding matrix. The formation of the Schottky barrier at the interface of the AgNW/TiO_2_ would also contribute to the transport of excited electrons from the AgNW interface to the surface of TiO_2_ due to the strong electric field [26]. Moreover, metal-semiconductor junction offers several other advantages, including a shorter e^−^–h^+^ pair diffusion distance and a more efficient photo-generated charge separation [27]. Ag nanowire-containing nanocomposite samples are expected to have a space charge region in the TiO_2_ side (near the semiconductor/metal junction). This, upon photoexcitation reaction, provides an additional force in-between holes and electrons, resulting in the holes and electrons moving in opposite directions when they are generated. This results in an overall comparatively lower e^−^–h^+^ pair recombination [13], which in turn results in the improved photocatalytic activity of the nanocomposite.

Under UV irradiation, photoexcited electrons from TiO_2_ nanostructures, in AgNW/TiO_2_/GO, move onto the graphene oxide surface (Figure 11c). Consequently, the graphene oxide sheet acts as an electron acceptor for TiO_2_, and thus increases the separation of electron–hole pairs and inhibits their recombination [28]. GO sheets may also act as an electron conductor that further promotes the electron–hole separation and prompts the rate of electron migration. Strong oxidizers, such as O_2_^−•^ and HO^•^, can be distributed along the GO surface, which would, in turn, increase the photo-degradation of pollutants [29].

## 4. Conclusions

Ag nanowire (AgNW)/TiO_2_ and AgNW/TiO_2_/GO nanocomposites were synthesized for water treatment purposes. The intention was to decorate Ag nanowires with TiO_2_ nanoparticles, using an in situ synthesis of TiO_2_ nanoparticles on Ag nanowires. In the AgNW/TiO_2_/GO nanocomposite sample, the idea was to have GO sheets dispersed within the composite. The photocatalytic activity of synthesized samples was assessed in Rhodamine B (RhB)-containing aqueous solution. RhB is a widely used dye in different industries. RhB removal from industrial wastewater is of prime importance for the environment. The following conclusions can be drawn from this investigation:-The proposed straightforward synthesis method resulted in the formation of Ag nanowires with an average length of 3 µm and an average diameter of 50 nm. XRD results also confirmed that TiO_2_ was successfully synthesized. In the AgNW/TiO_2_ sample, colonies of TiO_2_ nanoparticles were very well attached to Ag nanowires.-In AgNW/TiO_2_/GO nanocomposite samples, GO sheets were homogeneously dispersed within the composite mixture. Clusters of TiO_2_ nanoparticles were also found to be attached to GO sheets. FTIR results also confirmed the formation of atomic bonding at AgNW/TiO_2_ and GO/TiO_2_ junctions.-Results showed that synthesized AgNW/TiO_2_/GO nanocomposite has superior photocatalytic activity, compared to AgNW and AgNW/TiO_2_ samples. This has to do with higher light-harvesting due to the localized surface plasmon resonance (SPR) effect of Ag nanowires, lower e^−^–h^+^ pair diffusion distance, more effective photo-generated charge transfer, and more enhanced charge separation.

## Figures and Tables

**Figure 1 materials-14-00763-f001:**
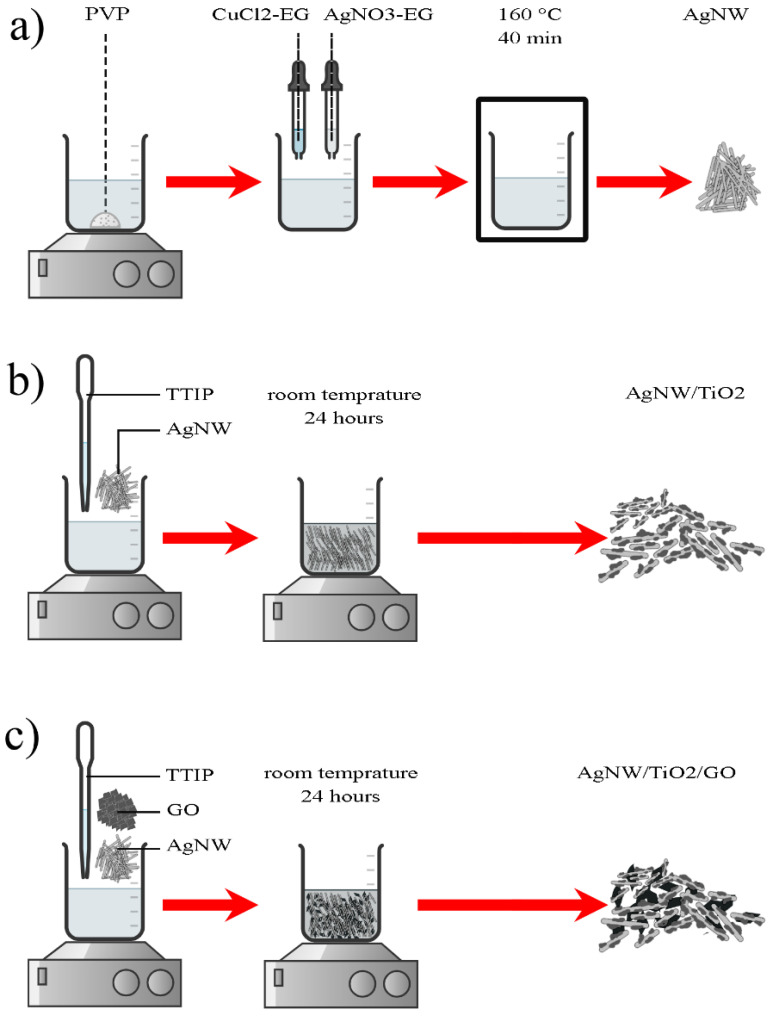
A schematic diagram describing the synthesis process of (**a**) AgNW, (**b**) AgNW/TiO_2_, and (**c**) AgNW/TiO_2_/GO samples.

**Figure 2 materials-14-00763-f002:**
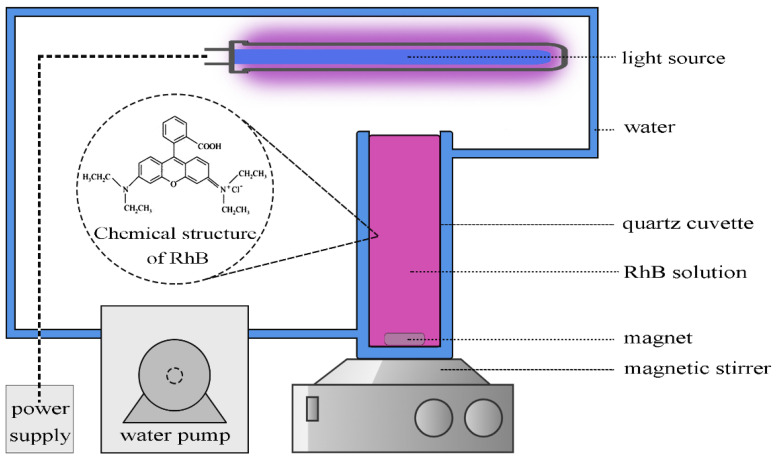
Schematic diagram of the photocatalytic reactor.

**Figure 3 materials-14-00763-f003:**
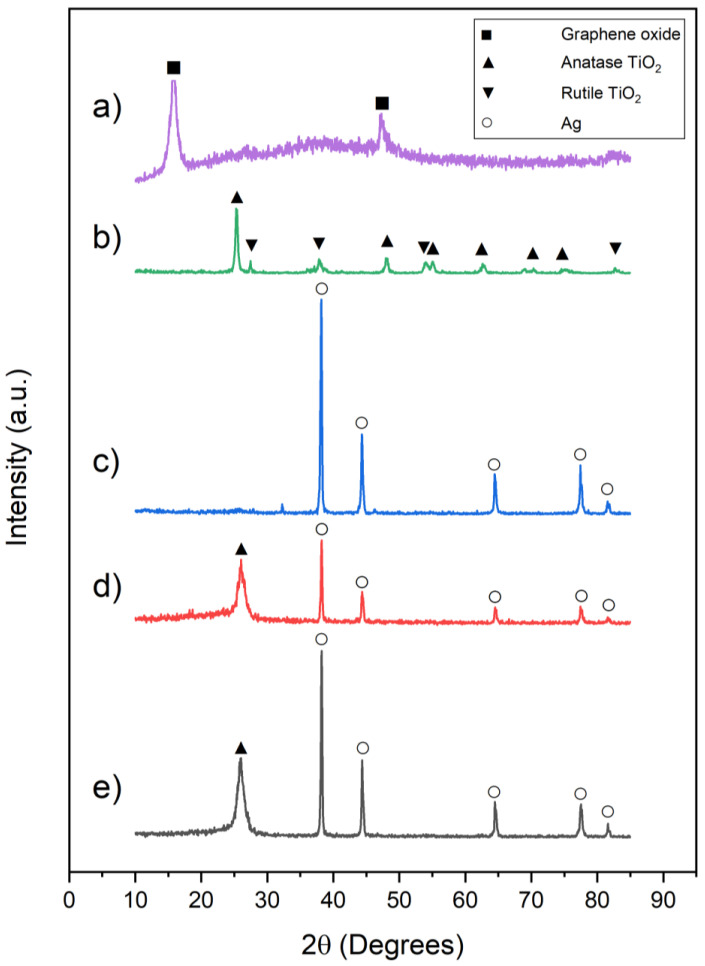
X-ray diffraction patterns of (**a**) GO, (**b**) P25 TiO_2_, (**c**) AgNW (**d**) AgNW/TiO_2_, and (**e**) AgNW/TiO_2_/GO samples.

**Figure 4 materials-14-00763-f004:**
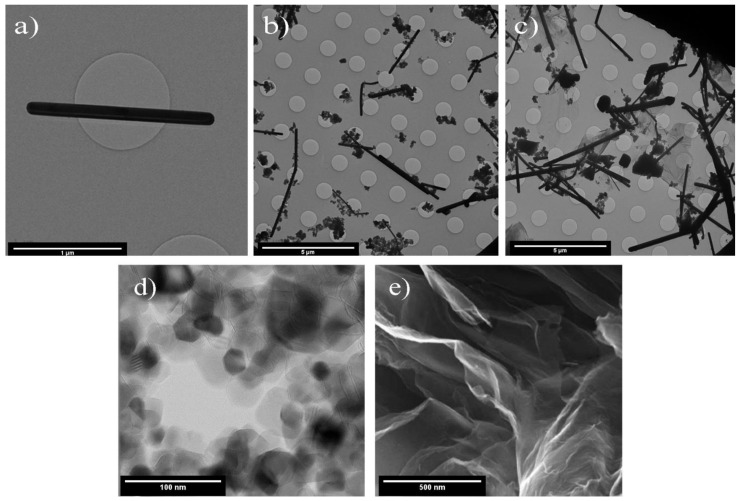
TEM images of (**a**) AgNW, (**b**) AgNW/TiO_2_, (**c**) AgNW/TiO_2_/GO (**d**) P25 TiO_2_, and (**e**) GO samples (image of GO is provided by the supplier, Pishgaman NanoMavad, Iran).

**Figure 5 materials-14-00763-f005:**
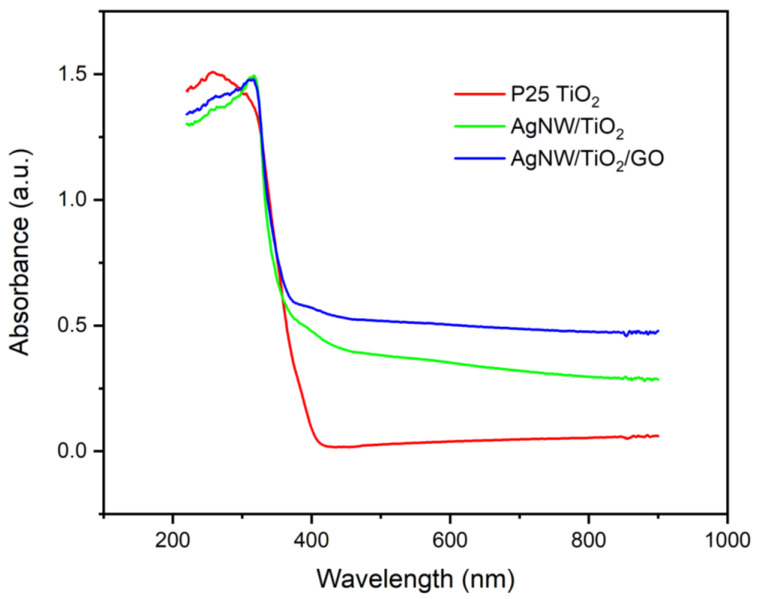
UV-VIS spectra of P25 TiO_2_, AgNW/TiO_2_, and AgNW/TiO_2_/GO nanocomposite samples.

**Figure 6 materials-14-00763-f006:**
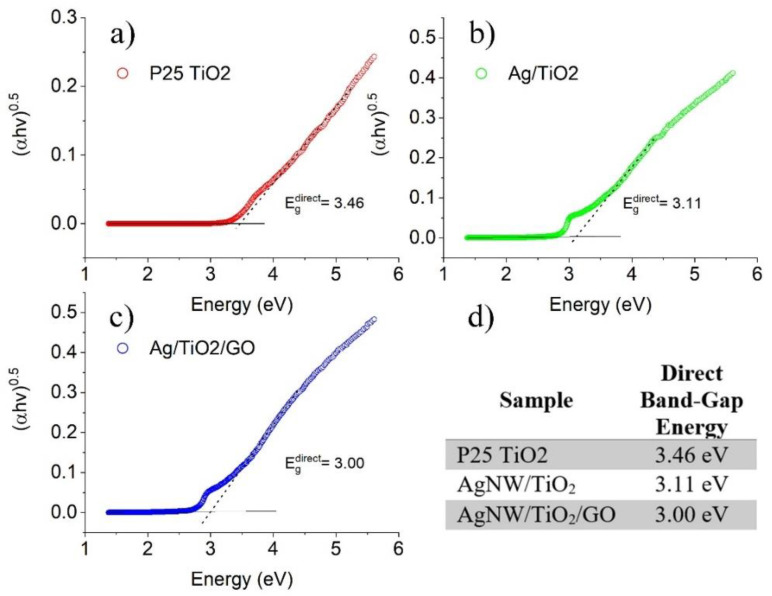
Tauc plot of the modified Kubelka–Munk (KM) function of (**a**) P25 TiO_2_, (**b**) AgNW/TiO_2_, (**c**) AgNW/TiO_2_/GO, and (**d**) calculated direct band gaps of each sample.

**Figure 7 materials-14-00763-f007:**
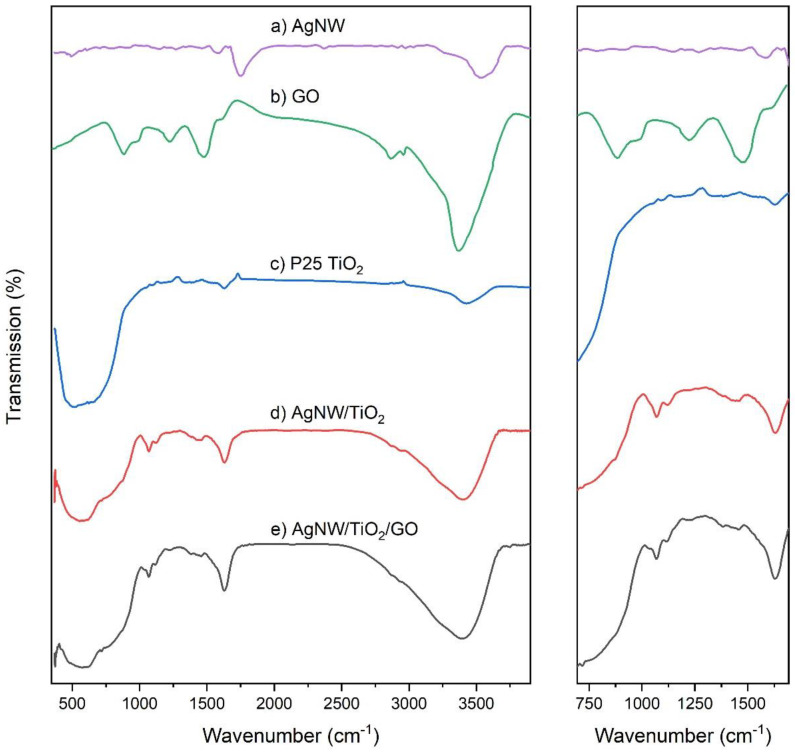
FTIR spectra of (**a**) AgNW, (**b**) GO, (**c**) P25 TiO_2_, (**d**) AgNW/TiO_2_, and (**e**) AgNW/TiO_2_/GO samples.

**Figure 8 materials-14-00763-f008:**
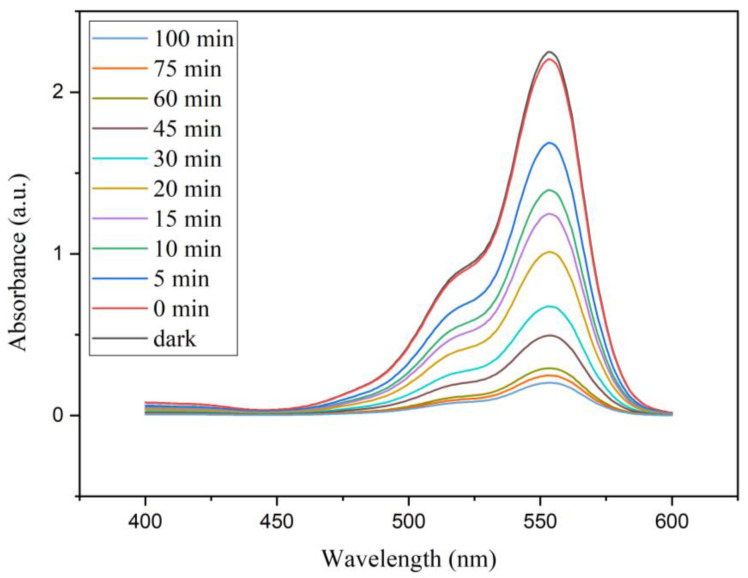
Time-dependent UV-visible absorption spectra of the RhB solution in the presence of AgNW/TiO_2_/GO nanocomposite samples.

**Figure 9 materials-14-00763-f009:**
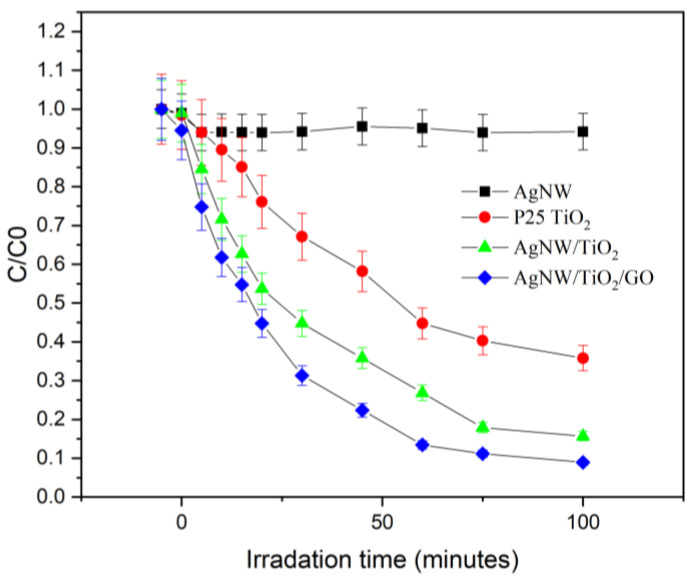
Photo-degradation of RhB for AgNW, P25 TiO_2_, AgNW/TiO_2_, and AgNW/TiO_2_/GO nanocomposite samples.

**Figure 10 materials-14-00763-f010:**
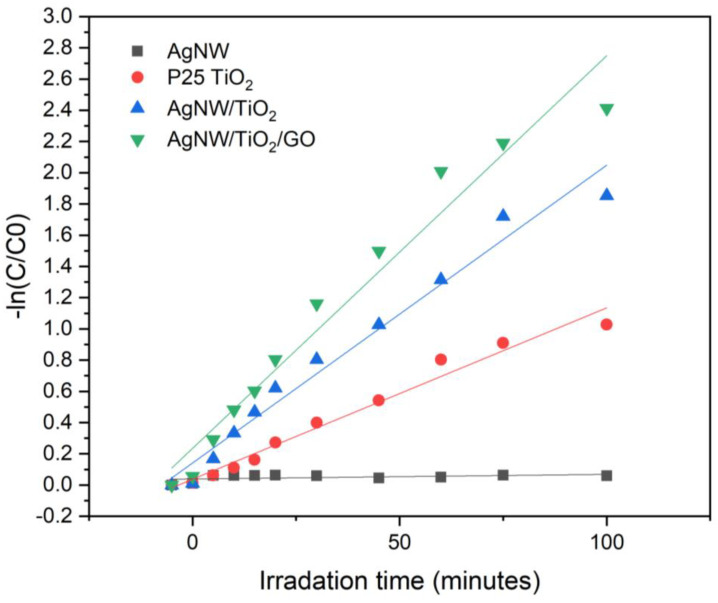
First-order kinetic curves of RhB degradation with AgNW, AgNW/TiO_2,_ and AgNW/TiO_2_/GO nanocomposite samples.

**Figure 11 materials-14-00763-f011:**
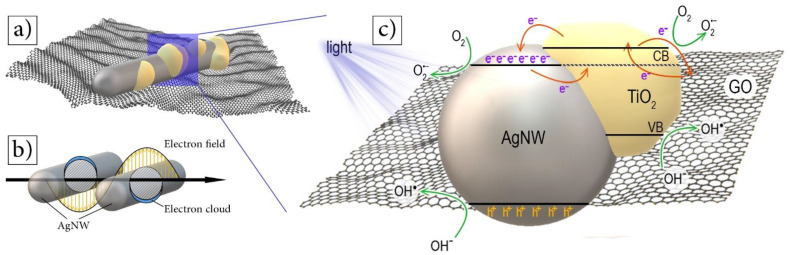
Schematic diagrams of (**a**) AgNW/TiO_2_/GO nanocomposite structure, (**b**) localized SPR of AgNWs when irradiated with visible or UV light, and (**c**) photocatalytic mechanism of AgNW/TiO_2_/GO nanocomposites.

**Table 1 materials-14-00763-t001:** Characteristics of commercial graphene oxide (GO).

Parameter	Values
Thickness	3.4–7 nm
Carbon Purity	~99%
Number of Layers	The average number of layers: 6–10
Surface Area (BET)	100–300 m^2^/g
Bulk Density	1 g/cc
Lateral dimension	10–50 μm

## Data Availability

The data presented in this study are available on request from the corresponding author.

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
