# Peer review of "Facile Synthesis of Ag Nanowire/TiO2 and Ag Nanowire/TiO2/GO Nanocomposites for Photocatalytic Degradation of Rhodamine B"

_materials, 2021, doi:10.3390/ma14040763_

Round 1
Reviewer 1 Report
The manuscript entitled, ‘Facile synthesis of Ag Nanowire/TiO2 and Ag Nanowire/TiO2/GO nanocomposites for photocatalytic degradation of Rhodamine B’ reported synthesis of heterogeneous catalyst for degradation of dye. This work infers some good findings though there are several loopholes that should be countered before publication. I am recommending some minor concerns for this work.
- Here Polyvinylpyrrolidone (PVP) was taken. What was the molecular weight of the Polyvinylpyrrolidone (PVP)?
- The concentration unit ‘mg/l’ should be corrected as ‘mg/L’.
- In Fig. 3, it will be better if the author mention the peaks in the image.
- What was average length or distribution of the length of the silver nanowires?
- Could the author reveal or predict what will be the differences in catalytic behaviors in between silver nanoparticles and nanowires?
- There are several articles in this area. I am recommending some of them for better literature review; 10.1016/j.surfin.2020.100611; 10.1016/j.ultsonch.2019.104797; 10.1016/j.apsusc.2018.08.133;
Author Response
Reviewer #1:
Dear Reviewer,
Many thanks for your comments. We tried to address all your comments in the manuscript. We put a lot of effort in addressing all your comments and we do believe that the quality of the manuscript has been significantly improved thanks to your inputs. We really appreciate that. Here, please find a detailed explanation of what we did to address your comments.
We hope you find this version publishable.
Should you have any further question, please do not hesitate to contact me.
Thanks again and regards,
Maryam Yazdan Mehr
Comments: 1) Here Polyvinylpyrrolidone (PVP) was taken. What was the molecular weight of the Polyvinylpyrrolidone (PVP)?
Average Mw was ~55,000 (it is now mentioned in the manuscript).
- The concentration unit ‘mg/l’ should be corrected as ‘mg/L’.
The manuscript was changed accordingly.
- In Fig. 3, it will be better if the author mention the peaks in the image.
Mentioned peaks are now also graphicly indexed in the Figure 3.
- What was average length or distribution of the length of the silver nanowires?
As mentioned in the manuscript (L 143-144) the average length for synthesized Ag nanowires are about 5 micro meters and an average diameter of 50 nm was measured.
- Could the author reveal or predict what will be the differences in catalytic behaviors in between silver nanoparticles and nanowires?
As described in (L 247-257), Spherical silver nanoparticles are expected to have lower photocatalytic efficiency, given that trapping sites on the surface of spherical particles further stimulates the e−/h+ recombination reactions. One-dimensional AgNWs decrease the charge localization over the surface, associated with reduced e−/h+ recombination. Reduced e−/h+ recombination has positive implications for the photocatalytic activities of nanocomposite samples. This improvement can be ascribed to the localized SPR effect of Ag nanowires. in this phenomenon, when AgNW is irradiated with light, The charge density is redistributed which creates a strong coulombic restoring force resulting in the oscillation of charge density in phase with the incident light
- There are several articles in this area. I am recommending some of them for better literature review; 10.1016/j.surfin.2020.100611; 10.1016/j.ultsonch.2019.104797; 10.1016/j.apsusc.2018.08.133;
Thanks for these interesting reads. These papers were studied and used in the revision of the manuscript and they are cited in the revised version.

Reviewer 2 Report
The presented paper fulfills the Journal Scopus. The research objectives and results are clearly stated. The authors motivated by cited literature carried out experiments and confirmed known knowledge. However, in my opinion, the presented paper does not show the novelty in the area. Moreover, the paper contributes nothing to the development of photochemistry and materials science disciplines. Additionally:
- Line 41-42, this sentence is meaningless.
- The goal is too general. The novelty should be more highlighted
- What is the role of the CuCl2 during the AG nanowire formation?
- Section 3.1. Figure 3 is described in the text as Figure 4. What are the reference numbers of XRD patterns?
- Figure 4, according to TEM measurements, the higher magnification would be more practical. What about the SEM data?
- What supplier do the authors mean in the description of Figure 5 according to the SEM images of the GO?
- The UV-ViS analysis of the AgNW should be shown to compare the obtained data.
- The calculation of the bandgap from the Tauc plot is wrongly designated.
- FTIR data should be filled by the GO and AgNW measurements.
- What about the stability of as formed heterostructures?
- Cycle photodegradation would improve the presented data.
- Authors claimed that at the surface of the materials, adsorbed are oxygen and hydroxyl radical. The analysis of the photocatalytic mechanism would improve the necessity of the heterostructures formation. (scavengers tests)
Recommendation Regarding This Manuscript: Reject
Author Response
Reviewer #2:
Dear Reviewer,
Many thanks for your comments. We tried to address all your comments in the manuscript. We put a lot of effort in addressing all your comments and we do believe that the quality of the manuscript has been significantly improved thanks to your inputs. We really appreciate that. Here, please find a detailed explanation of what we did to address your comments.
We hope you find this version publishable.
Should you have any further question, please do not hesitate to contact me.
Thanks again and regards,
Maryam Yazdan Mehr
Comments: 1) Line 41-42, this sentence is meaningless.
The sentence was revised.
- The goal is too general. The novelty should be more highlighted.
As per your suggestion, the last paragraph of the introduction was rewritten and the goal and novelty of the study were further clarified.
- What is the role of the CuCl2 during the AG nanowire formation?
It was found that a small amount of Cl− must be added to a polyol synthesis to provide electrostatic stabilization for the initially formed Ag seeds. in addition to this, the Cl− concentrations during CuCl2 mediated synthesis help reduce the concentration of free Ag+ ions in the solution through the formation of AgCl. Subsequently, it will slowly release the Ag+ ions. These facilitate the high-yield formation of the thermodynamically more stable multiply twinned Ag seeds that are required for wire length. Valency metal ions (Cu2+) are reduced by EG to low valence (Cu+) which in turn would react with and scavenged adsorbed atomic oxygen from the surface of AgNPs.
these mechanisms have been extensively studied by previous researchers. such as the following papers:
1- Johan, Mohd Rafie, et al. "Synthesis and growth mechanism of silver nanowires through different mediated agents (CuCl2 and NaCl) polyol process." Journal of Nanomaterials 2014 (2014).
2- Korte, Kylee E., Sara E. Skrabalak, and Younan Xia. "Rapid synthesis of silver nanowires through a CuCl-or CuCl 2-mediated polyol process." Journal of Materials Chemistry 18.4 (2008): 437-441.
- Section 3.1. Figure 3 is described in the text as Figure 4. What are the reference numbers of XRD patterns?
Figure number was corrected and the standard JCPDS card numbers were added to the manuscript text.
- Figure 4, according to TEM measurements, the higher magnification would be more practical. What about the SEM data?
SEM images we have does not add much to the content, as the size of features are very small. Presented TEM micrographs show the morphology rather well. SEM micrographs in this case are redundant.
- What supplier do the authors mean in the description of Figure 5 according to the SEM images of the GO?
The name of the supplier was added to Figure caption.
The UV-ViS analysis of the AgNW should be shown to compare the obtained data.
As AgNWs have shown no particular photocatalytic properties and have no band gap values, being metallic particles and not semiconductors, the Uv Vis measurements were deemed unnecessary and were not mentioned in the manuscript. Uv vis measurements were mainly done to calculate and compare band gap values.
The calculation of the bandgap from the Tauc plot is wrongly designated.
Many thanks for your comment. The calculation was redone and the correct results were added to the manuscript.
FTIR data should be filled by the GO and AgNW measurements.
FTIR data for GO and AgNW was added and the description was revised.
What about the stability of as formed heterostructures? Cycle photodegradation would improve the presented data. Authors claimed that at the surface of the materials, adsorbed are oxygen and hydroxyl radical. The analysis of the photocatalytic mechanism would improve the necessity of the heterostructures formation. (scavengers tests)
Many thanks for your suggestions. This manuscript is a first report on the observed photocatalytic characteristics in AgNW/TiO2. The manuscript is already almost 5000 word with 20 micrographs (images). Issues you have mentioned concering the the stability and cyclic photodegredation are indeed important. In fact, we are planning to combine mentioned experiments with electrochemical analyses (EIS, OCP, CV, …) to publish another manuscript. Adding all these data to this version makes this paper very long.

Reviewer 3 Report
In this work, "Facile synthesis of Ag nanowire/TiO2 and Ag nanowire/TiO2/GO nanocomposites for photocatalytic degradation of Rhodamine B", the authors demonstrated the photocatalytic characteristics of AgNW/TiO2 and AgNW/TiO2/GO nanocomposites. Based on the obtained results, the authors claimed that the synthesized AgNW/TiO2/GO nanocomposite has promising photocatalytic performance when it comes to the decomposition of RhB. Overall, this manuscript has a strong potential for a second review after applying the issues and addressing the shortcomings listed below:
1- The authors should polish/revise some grammatical mistakes and typos along the manuscript. To this end, I invite the authors to read their manuscript carefully and make the required changes where necessary.
2- As a general suggestion: Fix the length/width ratio of each Figure. The texts within the Figures seem extended along x direction.
3- In the Introduction section and along the manuscript, while discussing surface plasmon resonance concept and its possible application areas, the following work should also be considered and cited, to give a more general view to the possible readers of the work: [Monolithic metal dimer-on-film structure: new plasmonic properties introduced by the underlying metal, Nano Letters 20, 2087-2093 (2020)].
4- In Figure 6, what is the unit of absorbance? Is it a.u.?
5- Increase the resolution of Figures 7-8.
6- It seems we have two Figure 7. It should be fixed.
Author Response
Reviewer #3:
Dear Reviewer,
Many thanks for your comments. We tried to address all your comments in the manuscript. We put a lot of effort in addressing all your comments and we do believe that the quality of the manuscript has been significantly improved thanks to your inputs. We really appreciate that. Here, please find a detailed explanation of what we did to address your comments.
We hope you find this version publishable.
Should you have any further question, please do not hesitate to contact me.
Thanks again and regards,
Maryam Yazdan Mehr
Comments: 1) The authors should polish/revise some grammatical mistakes and typos along the manuscript. To this end, I invite the authors to read their manuscript carefully and make the required changes where necessary. As a general suggestion: Fix the length/width ratio of each Figure. The texts within the Figures seem extended along x direction.
Thanks for your comment. As per your suggestion, the manuscript was revised. Also, some o images (figures 3, 5, 6, 9, and 10) were changed and resized for a better visual apperence.
3- In the Introduction section and along with the manuscript, while discussing surface plasmon resonance concept and its possible application areas, the following work should also be considered and cited, to give a more general view to the possible readers of the work: [Monolithic metal dimer-on-film structure: new plasmonic properties introduced by the underlying metal, Nano Letters 20, 2087-2093 (2020)].
Thanks for your commet. This article was studied. It was very helpful and its now cited in the manuscript as Ref 24.
4- In Figure 6, what is the unit of absorbance? Is it a.u.?
The unit is arbitrary unit (a.u.) and was added to the image.
5- Increase the resolution of Figures 7-8.
As per your comment, the resolution of Figures 7 and 8 was enhanced
6- It seems we have two Figure 7. It should be fixed.
Figure numbers were corrected.

Round 2
Reviewer 2 Report
The changes made increase the value of the manuscript.
Reviewer 3 Report
In its current form, the revised manuscript is suitable for publication.